# The Dilemma of the Level of the Inferior Mesenteric Artery Ligation in the Treatment of Diverticular Disease: A Systematic Review of the Literature

**DOI:** 10.3390/jcm11040917

**Published:** 2022-02-10

**Authors:** Roberto Cirocchi, Giulio Mari, Bruno Amato, Giovanni Domenico Tebala, Georgi Popivanov, Stefano Avenia, Riccardo Nascimbeni

**Affiliations:** 1Department of Medicine and Surgery, University of Perugia, 06100 Perugia, Italy; cirocchiroberto@yahoo.it (R.C.); stefanoavenia1@gmail.com (S.A.); 2General Surgery Department, ASST Monza, Desio Hospital, Lombardia, 20833 Desio, Italy; 3Department of Clinical Medicine and Surgery, University Federico II of Naples, 80131 Naples, Italy; bramato@unina.it; 4Surgical Emergency Unit, John Radcliffe Hospital, Oxford University NHS Foundation Trust, Oxford OX3 7LE, UK; giovanni.tebala@ouh.nhs.uk; 5Department of Surgery, Military Medical Academy, “Sv. Georgi Sofiiski” 3 Str., 1606 Sofia, Bulgaria; gerasimpopivanov@rocketmail.com; 6Department of Molecular and Translational Medicine, University of Brescia, 25124 Brescia, Italy; riccardo.nascimbeni@unibs.it

**Keywords:** diverticular disease, vascular ligation, IMA, outcomes

## Abstract

Background and aim: Although sigmoidectomy is a well-standardized procedure for diverticular disease, there are still unclear areas related to the varying morphology and vascular supply of the sigmoid colon. The level of vascular ligation could affect the functional outcomes of patients operated on for diverticular disease. The aim of this review is to primarily evaluate sexual, urinary and defecatory function outcomes, as well as postoperative results, in patients who underwent surgery for diverticular disease, with or without inferior mesenteric artery (IMA) preservation. Materials and methods: The MEDLINE/PubMed, WOS and Scopus databases were interrogated. Comparative studies including patients who underwent sigmoidectomy for diverticular diseases were considered. Bowel function, genitourinary function, anastomotic leak, operation time, conversion to open surgery, anastomotic bleeding, bowel obstruction were the main items of interest. Results: Twelve studies were included in the review, three randomized and nine comparative studies. Bowel and genitourinary function are not differently affected by the level of vascular ligation. The site of ligation of IMA did not influence the rate of functional complications, anastomotic leak and bleeding. Of note, the preservation of IMA is associated with a higher conversion rate and longer operative time. Conclusions: Despite the heterogeneity of patient groups, and although the findings should be interpreted with caution, functional and clinical outcomes after sigmoidectomy for diverticular disease do not seem to be affected by the level of vascular ligation as long as the IMA is ligated far from its origin.

## 1. Introduction

Nowadays, sigmoidectomy and left hemicolectomy are the most commonly performed surgical procedures for symptomatic or complicated left-sided colonic diverticulosis. There is a general agreement that the sigmoid colon must be completely removed as it is the most common location of diverticulosis and diverticulitis in the Western world. Although sigmoidectomy is a well-standardized procedure, some technical variations exist according to the different morphologies and the vascularization of the sigmoid colon as follows: length of S-shaped loop between 11.9 and 91.1 cm, width of the sigmoid mesocolon between 4 and 11.5 cm [1], a highly variable pattern of sigmoid arteries—mainly divided into a common origin type—a separated origin or common trunk of the sigmoid arteries from the descending recto-sigmoid trunk [2] and the absence of the left colic artery [3].

In most cases, sigmoid resection is associated with an acceptable morbidity rate. Although anastomotic leak is the most feared complication, in the last decade a growing attention has been dedicated to the functional outcomes (genitourinary and bowel function above all) and quality of life (QoL). Although certain complications may be related to the patient’s general conditions, QoL and functional outcomes are mostly associated with the surgical technique. Therefore, several technical modifications have been proposed.

The inferior mesenteric artery (IMA) preservation is an attempt to reduce the anastomotic leak rate by preserving the blood supply of the rectal stump through the superior rectal artery. Although different surgical societies may not agree completely, 69.8% of members of the Consensus Conferences of EAES (European Association for Endoscopic Surgery) and SAGES (Society of American Gastrointestinal and Endoscopic Surgeons) agreed on the following recommendation: “Preservation of the inferior mesenteric artery should be considered to preserve the vascular supply to the anastomosis”. Moreover, they reported that IMA preservation was already their current practice. Twelve percent of surgeons agreed that this recommendation was likely to change their practice, while 18% disagreed [4]. Recently, the European Society of Coloproctology (ESCP) recommended the preservation of IMA “in cases where there is no suspicion of cancer to optimize the preservation of the vascularization and of the autonomic nerves” [5]. Similarly, the American Society of Colon and Rectal Surgery (ASCRS) stated that “some studies suggest that a mid-mesenteric dissection with preservation of the inferior mesenteric artery may decrease the incidence of the anastomotic leak; however, one meta-analysis failed to demonstrate a significant benefit with this approach” [6]. The WSES guidelines [7] and the multisocietary position statement of the Società Italiana di Chirurgia Colo-Rettale (SICCR), Società Italiana di Chirurgia d’Urgenza e del Trauma (SICUT), SIRM (Società Italiana di Radiologia Medica) and Associazione Italiana Gastroenterologi Ospedalieri (AIGO) [8] did not discuss this topic.

Another reason for a more peripheral dissection avoiding ligation of IMA, is the preservation of hypogastric nerves thus preventing urinary and sexual dysfunction. This is directly related to QoL. Surprisingly, only two small case series addressed the functional outcomes of sigmoidectomy for diverticular disease. Lesurtel et al. reported no significant impairment of the urinary and sexual function after central ligation of the IMA [9] and Forgione et al. reported similar results after peripheral dissection and preservation of IMA [10].

The aim of this systematic review is to investigate the clinical and functional outcomes in patients undergoing surgery for diverticular disease with either IMA preservation or ligation.

## 2. Materials and Methods

The review was performed according to the PRISMA (Preferred Reporting Items for Systematic Reviews and Meta-Analyses) 2020 statement [11].

### 2.1. The following Inclusion Criteria Were Used

-Randomized controlled studies or controlled observational studies, irrespective of the prospective or retrospective nature (only comparative studies);-Patients with acute colonic diverticulitis localized at left or sigmoid colon;-Ligation versus preservation of IMA.

### 2.2. The Exclusion Criteria Were

-Right or transverse colectomy;-Colon cancer associated with acute diverticulitis;-Patients undergoing only colostomy without colonic resection;-Patients undergoing segment colectomy or double-barrelled Mikulicz colostomy.

### 2.3. The Following Primary Functional Endpoints Were Assessed

-Bowel function;-Urinary function at 1 and 6 months after surgery, evaluated with the ICIQ (International Consultation on Incontinence Questionnaire) and IPSS (International Prostatic Symptoms Score);-Sexual function at 1 and 6 months after surgery, evaluated with the IIEF (International Index of Erectile Function) and FSFI (Female Sexual Function Index).

### 2.4. Secondary Endpoints Were

-Anastomotic leak rate;-Operation time-Conversion to open surgery;-Anastomotic bleeding;-Bowel obstruction.

A search on MEDLINE/PubMed, WOS (Web of Knowledge) and Scopus was performed. Variable associations of the following keywords were used: “inferior mesenteric artery”, “acute diverticulitis”, “preservation”, “sigmoidectomy”, “colectomy”, “sigmoid arteries” and “superior rectal artery”. Two authors (RC and GP) independently screened titles and abstracts; subsequently, they independently checked the full texts for potentially relevant studies. Similar articles from the same authors or group were highlighted and only the most recent were selected.

Furthermore, a Google Scholar search was performed, and Google Books was used for the analysis of the grey literature (https://books.google.com, accessed on 1 December 2021). Two Authors (RC and GP) performed data extraction from the selected articles based on the model of the “Cochrane Consumers and Communication Review Group” [12].

The quality of the included studies was evaluated by two authors (RC and GP) assessing the methodological quality according to the Cochrane Handbook for Systematic Reviews of Interventions for Randomized Controlled Trials (RCTs) [13] and the ROBINS-I (Risk of Bias In Non-randomized Studies of Interventions) assessment tool [14].

The extracted data were analysed by calculating the risk ratios (RR) for dichotomous variables and the weighted mean differences (WMD) for continuous variables. The analysis of RCTs was performed by the intention-to-treat analysis [15] and the Mantel–Haenszel method [16].

The results are shown in forest plot graphs (Figure 1). The I2 test was used to evaluate the heterogeneity of studies. A rate higher than 50% was considered as a significant heterogeneity. The meta-analysis was performed using the software Review Manager (RevMan) v 5.4.1 (Copenhagen: The Nordic Cochrane Centre, The Cochrane Collaboration, 2020).

## 3. Results

The electronic search strategy identified 1.063 articles. After the initial screening of titles and abstracts and removal of duplicated, 318 titles remained, of which 29 were evaluated as full text (Figure 1). Twelve studies met the inclusion criteria [17,18,19,20,21,22,23,24,25,26,27,28] and were included in the meta-analysis: 3 RCTs and 9 non-RCTs. Studies included a total of 2812 pooled patients as follows: 1612 underwent IMA ligation (IL) versus 1200 with IMA preservation (IP) (Table 1). Excluded studies are listed in Appendix A.

### 3.1. Characteristics of the Studies

The articles were published between 2001 and 2021, while the enrolment happened between 1982 and 2020. Eleven studies were performed in Europe and one in the USA (Table 1). Nine articles reported single-centre series, while three studies were multicentric.

Data about the timing of surgery were reported for 2516 patients (ten studies). Most of them had elective surgery (2.497 patients, 99.25%) and only 19 patients (0.75%) had emergency resection (Table 2).

The most common surgical approach was laparoscopy (87.41%, 2458 mini-invasive procedures, versus 12.59%, 354 open procedures) (Table 2).

In all studies, except the article by Lehmann et al. [22], the authors described the technique of IMA preservation (1.156 patients). The vascular control was achieved close to the colonic wall in 176 patients, 15.23%) [18,21,25], by ligation of the sigmoid vessels (287 patients, 19.12%) [24,26,27] and by Valdoni’s IMA skeletonization technique (262 patients, 22.66%) [17,23,28]. Most cases were extracted from the two articles of Posabella et coll., which included both techniques, dissection of the sigmoid vessels and peripheral dissection close to the colonic wall (497 patients, 42.99%) [19,20] (Table 2).

All the studies included patients with colorectal anastomosis with or without a diverting stoma.

### 3.2. Risk of Bias in Included Studies

The risk-of-bias assessment of the RCTs was performed according to the Cochrane Handbook for Systematic Reviews of Interventions (Figure 2). The trials of Mari et al. [26,27] and Masoni [18] et al. had a good quality in contrast to Tocchi et al. [28]. The recent RCT of Mari et al. [26] described exhaustively the modality of randomization and allocation concealment. The intraoperative randomization permits the exclusion of the patients with un-diagnosed mesosigmoid abscesses and reduces the rate of conversion for inflammatory and fibrotic alteration of the mesosigmoid. The old RCTs [18,28] reported the random sequence generation method, but they did not report the allocation method. The most common risk of bias in all RCTs is associated with the post-operative evaluation not being performed by independent evaluators. In all RCTs, the incomplete outcome data (attrition) bias was low (actually, none of the patients was lost during the studies). Only the study of Masoni et al. [18] reported the protocol approval from the Ethical Committee and the registration on ClinicalTrials.gov, accessed on 1 December 2021 (NCT01326052).

The risk-of-bias judgement of the CCTs was performed according to the ROBINS-I tool and confirmed a risk of bias (Figure 3). Regarding the bias due to confounding factor, five studies did not report the severity of acute diverticulitis [17,22,23,25,26]; it was therefore impossible to evaluate the presence of an unequal distribution of this additional risk factor between the study groups; one study described the same rate of severity of acute diverticulitis reaching a low risk of bias [27]. On the contrary, three other studies were assessed with moderate risk of overall bias [19,20,21]. In both studies by Posabella, the rate of recurrent diverticulitis with covered perforation was higher in the IP group [19,20]. In the study of Borchert, the characteristics of patients based on the Hansen–Stock classification of acute diverticulitis are very different, with the highest severity being reported in the IP group [21]. Analysing the bias in the selection of participants, two CCTs did not report the timing of surgical treatment (emergency vs. elective) [17,21] and five CCTs excluded patients who underwent emergency surgery [19,20,23,24,25]. Only two studies have a moderate risk of selection bias for the inclusion of a relatively small number of patients in the emergency setting [22,26]. Regarding the bias in the classification of interventions, the majority of trials had a low risk, except for three papers that did not report the level of IMA ligation [17,22,23] or the technique of IMA-preservation [23], and two articles in which the authors included patients who underwent peripheral dissection on the sigmoid vessels or close to the colon [19,20]. In these cohort studies, it was impossible to evaluate the risk of allocation bias. The risk of bias in the outcome evaluation was estimated as low in all studies, except for one study of Posabella et al. who evaluated the functional outcomes with a questionnaire administered in a short period on a large cohort of patients who received surgery between 2004 and 2017 [20]. As regards the bias due to missing data, only one study had a moderate risk while the others had a low risk. Posabella et al. reported a high number of patients not returning the questionnaire on genitourinary function, with only 442 questionnaires out of 1213 (36.44%) being returned. Of them, 118 (9.73%) were excluded for different reasons, leaving only 324 questionnaires (26.71%) available for the evaluation. The risk of bias in the selection of reported results was considered low in all studies [19].

### 3.3. Primary Outcomes

#### 3.3.1. Bowel Function

This outcome was evaluated using the Gastrointestinal Quality of Life Index (GIQLI) [29] by Mari et al. [27], and the Jorge–Wexner incontinence score [30] and the Agachan–Wexner constipation score [31] by Masoni et al. [18]. The other studies did not report this outcome. The non-uniformity of the evaluation scales prevented a possible meta-analysis of those two studies. In the RCT of Mari et al. the GIQLI reported a lower score in the first month (*p* > 0.05) compared to base line and a return to the preoperative level at six months (*p* > 0.05) in the IMA-ligation group. On the contrary, Masoni et al. reported a lower rate of defecation disorders (constipation, sensation of incomplete defecation and unsuccessful defecation) in the IMA-preservation group compared to the ligation group at six months.

#### 3.3.2. Urinary Function

This outcome was evaluated through the AUASI (American Urological Association Symptom Index) [32] by Posabella et al. [19] and through the ICIQ (International Consultation on Incontinence Questionnaire) [33] and IPSS (International Prostatic Symptoms Score) [34] by Mari et al. [27]. Both studies showed no difference between the two groups in long-term results.

#### 3.3.3. Sexual Function

This outcome at one and six months after surgery was evaluated with the IIEF (International Index of Erectile Function) [35] and FSFI (Female Sexual Function Index) [35] in the study of Mari et al. [24,27], reporting overlapping results in the two groups (*p* > 0.05). The same results were reported by Posabella et al. in the IIEF analysis (*p* = 0.674). [19]. The FSFI score evaluated by Mari et al. did not show any difference between the two groups (*p* > 0.05) [27].

### 3.4. Secondary Endpoints

#### 3.4.1. Clinical Anastomotic Leak

This outcome was reported by 10 studies (two RCTs, [27,28] and eight CCTs, (2383 patients) [17,20,21,22,23,24,25,26]. The leak rate was lower in patients who had IMA preservation (2.09%, 30/1436) compared to those who had IMA ligation (3.91%, 37/947), but this difference is not statistically significant (RR 0.73, 95% CI 0.35 to 1.53, I2 = 40%). Similarly, the subgroup analysis of RCTs (RR 0.34, 95% CI 0.04 to 3.09) and CCTs (RR 0.86, 95% CI 0.39 to 1.91) did not find any difference (Figure 4).

Furthermore, the sensitivity analysis of the subgroup of patients who were operated on laparoscopically yielded the same results (RR 0.72, 95% CI 0.38 to 1.37, I2 = 0%) (Figure 5).

Additionally, the subgroup analyses of IMA preservation vs. high ligation and IMA preservation vs. low ligation had similar findings (RR 0.55, 95% CI 0.12 to 2.55, and RR 0.64, 95% CI 0.19 to 2.16, respectively) (Figure 6a,b).

#### 3.4.2. Rectal Bleeding

This outcome was reported in six studies (1.656 patients) [17,20,24,25,26,27]. Post-operative rectal bleeding happened more frequently in patients who underwent IMA preservation although not significantly, with results as follows: 2.66% vs. 1.33% (RR 1.98, 95% CI 0.96–4.10) (Figure 7).

#### 3.4.3. Operative Time

Operative time was significantly longer in the IMA preservation group (MD 21.96, 95% CI 4.52–39.39), (3 studies, 341 patients) [18,26,27] (Figure 8).

The subgroup analysis of the different IMA preservation reported the same trend at favour of the IMA preservation group; however, no results were statistically significant (Figure 9).

#### 3.4.4. Conversion to Open Surgery

This outcome was obviously evaluated only in the laparoscopic series (eight studies, 1990 patients) [18,20,21,23,24,25,26,27]. Conversion rate was significantly higher in the IMA preservation group, as follows: 7.64% vs. 4.29% (MD 1.65, 95% CI 1.10–2.47) (Figure 10).

Furthermore, a subgroup analysis was performed to evaluate how the different techniques of AMI-preserving affected the conversion rate. We noticed a higher although not statistically significant, laparotomic conversion rate associated with all the different techniques for IMA preservation (Figure 11).

## 4. Discussion

The present analysis revealed that IMA preservation does not significantly affect post-operative bowel, sexual and urinary functions in patients undergoing surgery for diverticular disease. Anastomotic leak rate and rectal bleeding rate do not seem to be associated with IMA preservation or ligation [36].

The debate about the level of vascular ligation during elective sigmoidectomy for diverticular disease has changed over the past few years [2,3]. The initial position was to apply the surgical steps usually performed in sigmoidectomy for malignant disease, due to the risk of a misdiagnosed colonic cancer [22]. Furthermore, it was considered good practice to always perform the same operation in order to improve the surgeon’s confidence with the manoeuvre, irrespective of the nature of the disease. The crucial technical point was not to dissect the IMA close to the aorta to reduce the risk of injuring the superior hypogastric plexus [28]. However, a rigorous and prospective evaluation of the genitourinary function in patients operated on for left-sided DD with a high-tie of IMA has never been conducted, even if sexual complications have been reported [26,27].

In the last ten years, several studies investigated the impact of the level of vascular ligation on postoperative functional outcomes [18,19,27]. Due to the increasing number of patients undergoing elective surgery for DD a growing attention has been paid to QoL [7,36]. The unspoken “motto” implied in QoL was: “if you are not able to improve it, at least do not worsen it”. Hence, the increasingly accepted tendency to move the level of vascular control more distally, towards the colonic wall [19].

As emerged from our analysis, the distal ligation of IMA can increase the conversion to open surgery. The distal ligation implies an intra-mesocolic dissection rather than a smooth dissection through the avascular planes. In the case of chronically inflamed thick and fibrotic mesocolon, or even worse in the case of an intra-mesocolic abscess, the risk of intraoperative bleeding and misconception of the anatomical landmarks could lead to a higher rate of conversion [27]. However, it must be emphasized that this outcome should be interpreted with caution, as it is not always clear if the conversion happened before, during or after the vascular preparation. Moreover, it also depends on the technical skills of the operator.

When the dissection is performed away from the aortic plane, the risk of injuring the superior hypogastric plexus and the hypogastric nerves is significantly reduced, with a possible sparing effect on the genitourinary function [37,38]. Our findings, however, confirm a substantial equivalence of the two types of vascular ligation. Regarding bowel function, some authors underline the importance of the combination between the nerve fibres of the hypogastric plexus and those running along with the IMA to the rectum and distal sigmoid colon [39]. In this setting, a complete preservation of the IMA could spare the parasympathetic component involved in bowel function, thus resulting in better functional outcomes. Although interesting from the anatomical-physiological point of view, this hypothesis does not seem to find confirmation in our review. Only one RCT, in fact, reported a lower incidence of defecation disorders at 6 months in the IMA preservation group [18].

Focusing on the problem of anastomotic leak, theoretically, IMA preservation should guarantee a better blood supply of the rectal stump, but sound evidence is lacking. Furthermore, from a practical point of view, the introduction of new anastomotic perfusion confirmation techniques, such as the indocyanine green fluorescence, makes future studies on this issue virtually useless [40].

The efforts to establish prospective multicentre registries on diverticular disease could pave the way for a new type of evidence on the treatment of diverticular disease [41]. Given the difficulty in carrying out RCTs without suffering the bias of the samples’ heterogeneity and without incurring in ethical issues, the availability of follow-up data of patients operated on for diverticular disease will provide precious information for the decision-making process.

The limitations of this study are related to the high risk of selection, performance and detection bias of the included RCT, and to the relatively small sample size of the studies evaluating the functional outcomes.

## 5. Conclusions

The site of ligation of the IMA does not influence the rate of functional complications, anastomotic leak rate and bleeding rate. The preservation of IMA may be, however, associated with a higher conversion rate and longer operative time. Although this finding should be interpreted with caution, it suggests a careful approach to the intra-mesocolic dissection. The crucial technical point is the dissection of the IMA 1–2 cm above the aorta to preserve the superior hypogastric plexus. A multicentre registry of diverticular disease may provide a piece of new and more reliable evidence regarding the technical aspects of the surgical treatment.

## Figures and Tables

**Figure 1 jcm-11-00917-f001:**
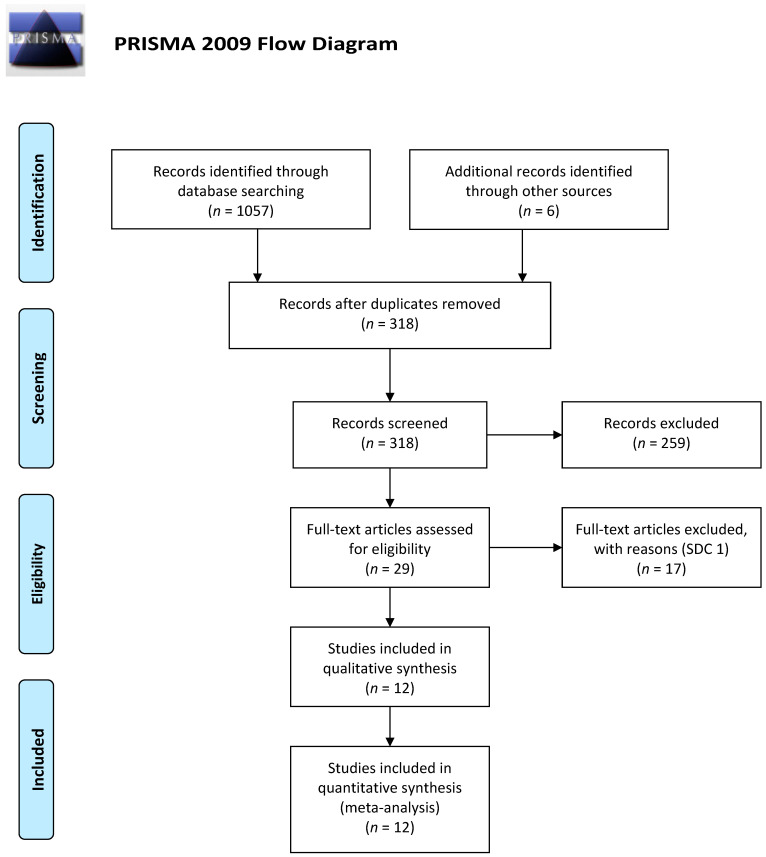
Prisma flow diagram of selected studies.

**Figure 2 jcm-11-00917-f002:**
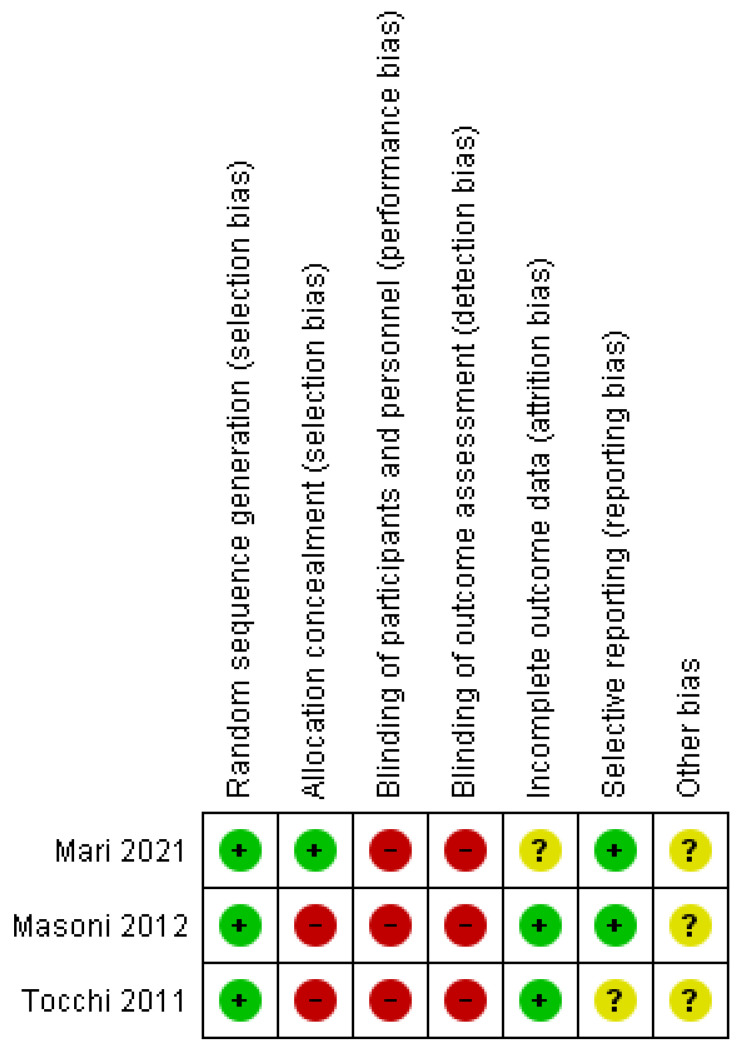
The risk-of-bias judgment of the RCTs, performed according to Cochrane Handbook for Systematic Reviews of Interventions: review authors’ judgements about each risk of bias item for each included study.

**Figure 3 jcm-11-00917-f003:**
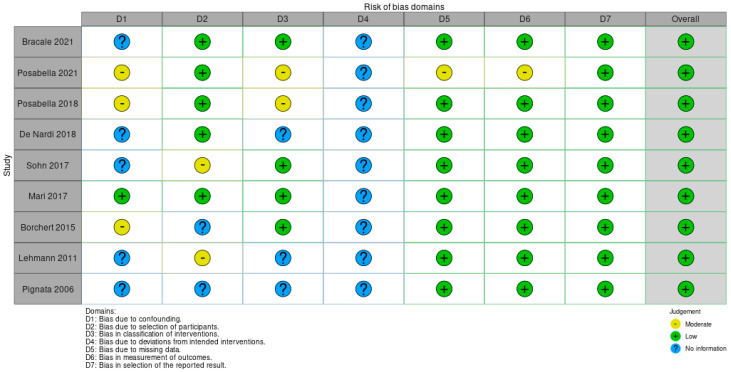
The risk-of-bias judgment of the CCTs, performed according to ROBINS-I tool, risk-of-bias identified.

**Figure 4 jcm-11-00917-f004:**
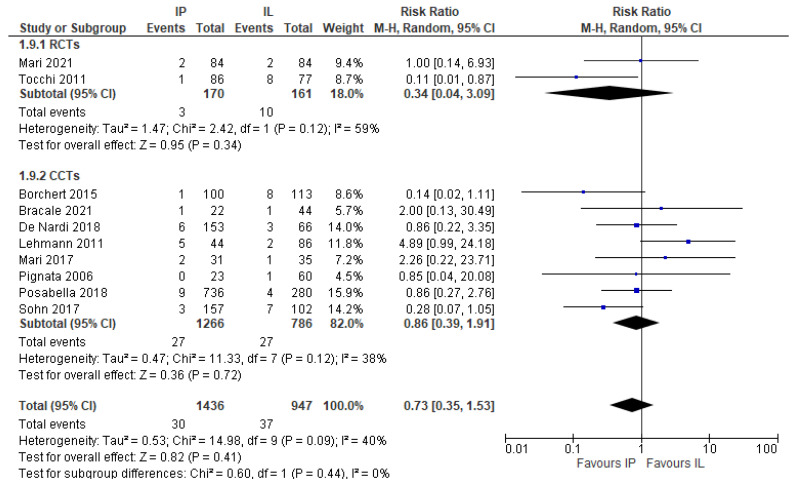
Anastomotic leak risk ratio.

**Figure 5 jcm-11-00917-f005:**
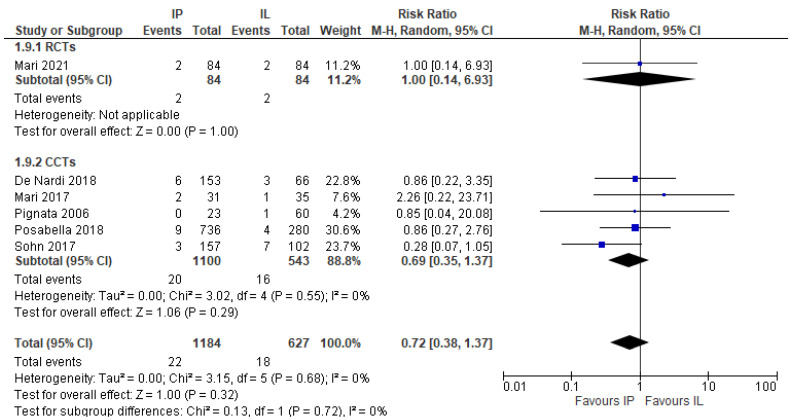
Anastomotic leak risk ratio in the laparoscopic sub-group.

**Figure 6 jcm-11-00917-f006:**
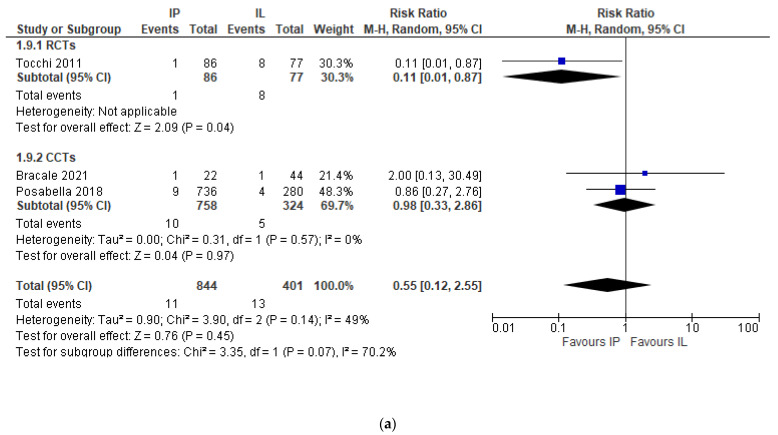
(**a**) Anastomotic leak, risk ratio, IMA preservation vs. IMA high ligation. (**b**) Anastomotic leak, risk ratio, IMA preservation vs. IMA low ligation.

**Figure 7 jcm-11-00917-f007:**
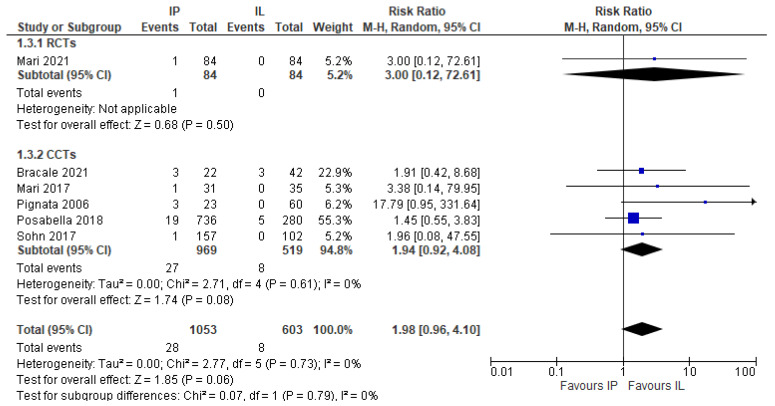
Post-operative rectal bleeding, risk ratio.

**Figure 8 jcm-11-00917-f008:**
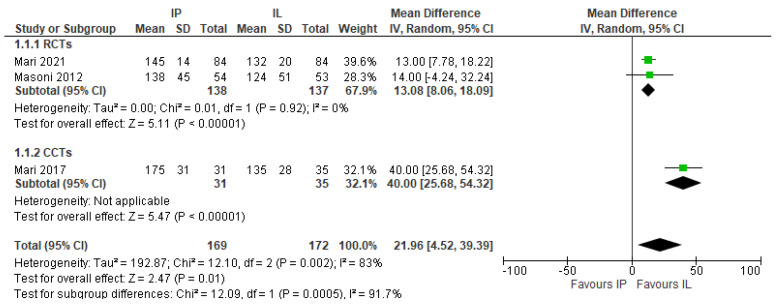
Operative time, risk ratio.

**Figure 9 jcm-11-00917-f009:**
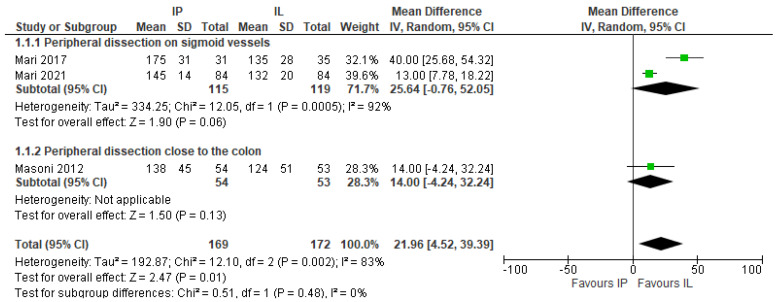
Operative time. Risk ratio for different IMA preservation techniques.

**Figure 10 jcm-11-00917-f010:**
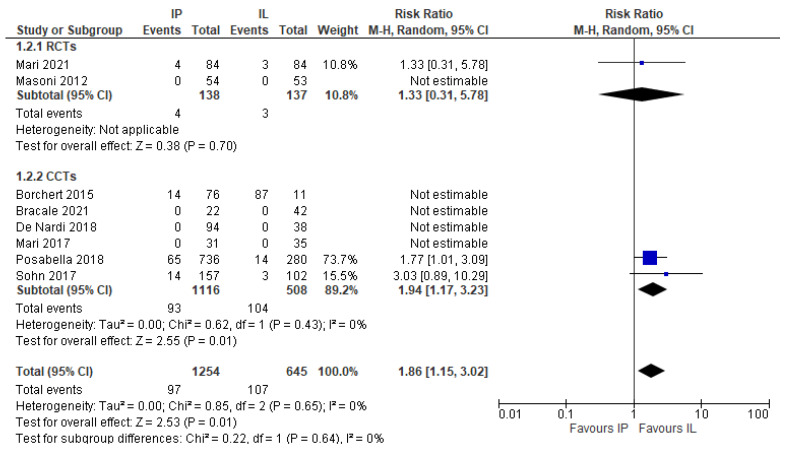
Conversion rate, risk ratio in the laparoscopic subgroup.

**Figure 11 jcm-11-00917-f011:**
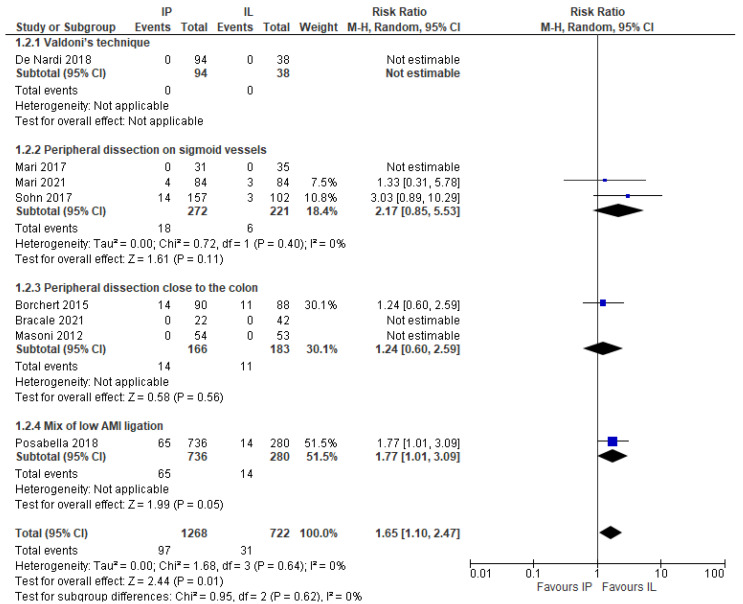
Conversion rate, risk ratio in the laparoscopic subgroup according to different technique for IMA preservation.

**Table 1 jcm-11-00917-t001:** Characteristics of studies included in the meta-analysis.

Authors and Year of Publication	Type of Study	Number of Centers	Enrollment Period	Nation	No. of Patients Enrolled	IMA Ligation	IMA Preserving
Mari 2021	RCT	Multicenter	2016–2019	Italy	168	84	84
Bracale 2021	RCCS	Monocenter	2018–2020	Italy	64	42	22
Posabella 2021	RCCS	Monocenter	2004–2017	Switzerland	324	107	217
Posabella 2018	RCCS	Monocenter	2004–2014	Switzerland	1016	736	280
De Nardi 2018	RCCS	Monocenter	2006–2012	Italy	219	66	153
Sohn 2017	RCCS	Multicenter	2002–2015	Germany	259	157	102
Mari 2017	PCCS	Multicenter	2015–2016	Italy	66	31	35
Borchert 2015	RCCS	Monocenter	2006–2008	Germany	213	113	100
Masoni 2012	RCT	Monocenter	2004–2010	Italy	107	53	54
Lehmann 2011	RCCS	Monocenter	2002–2009	USA	130	86	44
Pignata 2006	RCCS	Monocenter	2000–2005	Italy	83	60	23
Tocchi 2001	RCT	Monocenter	1982–1996	Italy	163	77	86
Total					2812	1612	1200

RCT: randomized controlled trial; PCCS: prospective case–control study; RCCS: retrospective case–control study; IMA: inferior mesenteric artery.

**Table 2 jcm-11-00917-t002:** Characteristics of surgical treatment in included studies.

Authors and Year of Publication		Access to Surgical Treatment (No. of Patients)		Setting of Surgical Treatment (No. of Patients) (%)	Technique of IMA-Preservation	Level of IMA Ligation
		Open	Laparoscopy			
Mari 2021		0	168 (100%)	168 elective (100%)	Peripheral dissection on sigmoid vessels	Low
Bracale 2021		2 (3.23%)	62 (96.77%)	64 elective (100%)	Peripheral dissection close to the colon	High
Posabella	2021	0	324 (100%)	324 elective (100%)	Peripheral dissection on sigmoid vessels or close to the colon	High
	2018	0	1016 (100%)			High
De Nardi 2018		87 (39.73%)	132 (60.27%)	219 elective (100%)	Valdoni’s technique	NR
Mari 2017		0	66 (100%)	66 elective (100%)	Peripheral dissection on sigmoid vessels	Low
Sohn 2017		0	259 (100%)	257 elective (99.23%)—2 emergency (0.67%)	Peripheral dissection on sigmoid vessels	Low
Borchert 2015		25 (11.74%)	188 (88.26%)	213 elective (100%)	Peripheral dissection close to the colon	High or low
Masoni 2012		0	107 (100%)	107 elective (100%)	Peripheral dissection close to the colon	High
Lehmann 2011		77 (59.23%)	53 (60.77%)	113 elective (86.92%)—17 emergency (13.08%)	NR	NR
Pignata 2006		0	83 (100%)	NR	Valdoni’s technique	NR
Tocchi 2001		163 (100%)	0	163 elective (100%)	Valdoni’s technique	High

Close colon: peripheral dissection close to the colon; Sigmoid vessels: peripheral dissection on sigmoid vessels; Valdoni’s technique of IMA skeletonization (“*the adventitia of the inferior mesenteric artery is entered on the posterior aspect of the vessel. The dissection continues in this plane, and all the branches directed to the left and sigmoid colon are divided*”); High: ligation of the inferior mesenteric artery at origin from aorta; Low: ligation of the inferior mesenteric artery after the left colic artery.

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
