# Peer review of "The Dilemma of the Level of the Inferior Mesenteric Artery Ligation in the Treatment of Diverticular Disease: A Systematic Review of the Literature"

_jcm, 2022, doi:10.3390/jcm11040917_

Round 1
Reviewer 1 Report
Paying attention to the following references can be useful
Mari G, Montroni I, Origi M, Crippa J, Achilli P, Costanzi A, Ferrari G, Ugolini G, Maggioni D, AIMS Academy Clinical Research Network
Functional and Short-term Outcomes in Elective Laparoscopic Colectomy for Symptomatic Diverticular Disease With Either Low Ligation or Inferior Mesenteric Artery Preservation: A Randomized Trial.
Surgical laparoscopy, endoscopy & percutaneous techniques. 2020
Kruszewski WJ, Szajewski M, Ciesielski M, Buczek T, Kawecki K, Walczak J
Level of inferior mesenteric artery ligation does not affect rectal cancer treatment outcomes despite better cancer-specific survival after low ligation-randomized trial results.
Colorectal Dis. 2021
Mari G, Crippa J, Costanzi A, Mazzola M, Magistro C, Ferrari G, Maggioni D
Genito-Urinary Function and Quality of Life after Elective Totally Laparoscopic Sigmoidectomy after at Least One Episode of Complicated Diverticular Disease According to Two Different Vascular Approaches: the IMA Low Ligation or the IMA Preservation.
Chirurgia (Bucharest, Romania : 1990). 2017
Cirocchi R, Popivanov G, Binda GA, Henry BM, Tomaszewski KA, Davies RJ, Di Saverio S
Sigmoid resection for diverticular disease - to ligate or to preserve the inferior mesenteric artery? Results of a systematic review and meta-analysis.
Colorectal Dis. 2019
Cirocchi R, Trastulli S, Farinella E, Desiderio J, Listorti C, Parisi A, Noya G, Boselli C
Is inferior mesenteric artery ligation during sigmoid colectomy for diverticular disease associated with increased anastomotic leakage? A meta-analysis of randomized and non-randomized clinical trials.
Colorectal Dis. 2012
Borchert DH, Schachtebeck M, Schoepe J, Federlein M, Bunse J, Gellert K, Burghardt J
Observational study on preservation of the superior rectal artery in sigmoid resection for diverticular disease.
International journal of surgery (London, England). 2015
Tsujisaka Y, Kitai T, Kaji S, Kinoshita M, Nakamura K, Koyama T, Furukawa Y
Sole Obstruction of the Inferior Mesenteric Artery With Acute Aortic Dissection Causing Critical Mesenteric Ischemia.
JACC. Case reports. 2020
De Nardi P, Gazzetta P
Does inferior mesenteric artery ligation affect outcome in elective colonic resection for diverticular disease?
ANZ journal of surgery. 2018
Cirocchi R, Trastulli S, Farinella E, Desiderio J, Vettoretto N, Parisi A, Boselli C, Noya G
High tie versus low tie of the inferior mesenteric artery in colorectal cancer: a RCT is needed.
Surgical oncology. 2012
Jolivet M, Trilling B, Sage PY, Boussat B, Girard E, Faucheron JL
Prospective evaluation of functional outcomes after laparoscopic sigmoidectomy with high tie of the inferior mesenteric artery for diverticular disease in consecutive male patients.
Techniques in coloproctology. 2019
Kihara K, Horie H, Miyatani K, Endo M, Matsunaga T, Yamamoto M, Yata S, Tokuyasu N, Sakamoto T, Fujiwara Y
A rare case of sigmoid colon cancer in which the lower limbs received collateral blood flow from the inferior mesenteric artery owing to peripheral artery disease.
Surgical case reports. 2021
Bradley S, Quenzer F, Wittler M
Ruptured Visceral Artery Aneurysms: A Deadly Cause of Epigastric Pain.
Clinical practice and cases in emergency medicine. 2019
Trilling B, Riboud R, Abba J, Girard E, Faucheron JL
Energy vessel sealing systems versus mechanical ligature of the inferior mesenteric artery in laparoscopic sigmoidectomy.
International journal of colorectal disease. 2016
Lee KH, Kim JS, Kim JY
Feasibility and oncologic safety of low ligation of inferior mesenteric artery with D3 dissection in cT3N0M0 sigmoid colon cancer.
Annals of surgical treatment and research. 2018
Charan I, Kapoor A, Singhal MK, Jagawat N, Bhavsar D, Jain V, Kumar V, Kumar HS
High Ligation of Inferior Mesenteric Artery in Left Colonic and Rectal Cancers: Lymph Node Yield and Survival Benefit.
The Indian journal of surgery. 2014
Jonnada PK, Karunakaran M, Rao D
Outcomes of level of ligation of inferior mesenteric artery in colorectal cancer: a systematic review and meta-analysis.
Future oncology (London, England). 2021
Yoo SJ, Ku MJ, Cho SS, Yoon SP
A case of the inferior mesenteric artery arising from the superior mesenteric artery in a Korean woman.
Journal of Korean medical science. 2011
Dimitriou N, Felekouras E, Karavokyros I, Pikoulis E, Vergadis C, Nonni A, Griniatsos J
High versus low ligation of inferior mesenteric vessels in rectal cancer surgery: A retrospective cohort study.
Journal of B.U.ON. : official journal of the Balkan Union of Oncology. 2018
Singh D, Luo J, Liu XT, Ma Z, Cheng H, Yu Y, Yang L, Zhou ZG
The long-term survival benefits of high and low ligation of inferior mesenteric artery in colorectal cancer surgery: A review and meta-analysis.
Medicine. 2017
Parma P, Samuelli A, Dall'Oglio B
[Laparoscopic repair of enterovesical fistula in patient affected by left megaureter].
Urologia. 2013
Masoni L, Mari FS, Nigri G, Favi F, Gasparrini M, Dall'Oglio A, Pindozzi F, Pancaldi A, Brescia A
Preservation of the inferior mesenteric artery via laparoscopic sigmoid colectomy performed for diverticular disease: real benefit or technical challenge: a randomized controlled clinical trial.
Surgical endoscopy. 2012
Cirocchi R, Farinella E, Trastulli S, Desiderio J, Di Rocco G, Covarelli P, Santoro A, Giustozzi G, Redler A, Avenia N, Rulli A, Noya G, Boselli C
High tie versus low tie of the inferior mesenteric artery: a protocol for a systematic review.
World journal of surgical oncology. 2011
Papagrigoriadis S
Differences in early outcomes after open or laparoscopic surgery: what is the evidence?
Digestive diseases (Basel, Switzerland). 2012
Campbell A, Macdonald A, Oliphant R, Russell D, Fogg QA
Neurovasculature of high and low tie ligation of the inferior mesenteric artery.
Surgical and radiologic anatomy : SRA. 2018
Fujii S, Ishibe A, Ota M, Watanabe K, Watanabe J, Kunisaki C, Endo I
Randomized clinical trial of high versus low inferior mesenteric artery ligation during anterior resection for rectal cancer.
BJS open. 2018
Brenes RA, Panait L, Abbas HM, Tapias L, Tripodi G, Ajemian MS, Macaron SH
Prevention of type II endoleak by laparoscopic inferior mesenteric artery ligation.
Surgical innovation. 2012
Lotun K, Shetty R, Topaz O
Atherosclerotic inferior mesenteric artery stenosis resulting in large intestinal hypoperfusion: a paradigm shift in the diagnosis and management of symptomatic chronic mesenteric ischemia.
Catheter Cardiovasc Interv. 2012
Elkaouini A, Berrajaa S, Aabdi M, Merbouh M, El Mezzeoui S, El Aidouni G, Bkiyar H, Jabi R, Bouziane M, Housni B
Inferior mesenteric artery aneurysm revealed by massif rectal bleeding, case report.
Annals of medicine and surgery (2012). 2021
Polistena A, Cavallaro G, D'Ermo G, Paliotta A, Crocetti D, Rosato L, De Toma G
Clinical and surgical aspects of high and low ligation of inferior mesenteric artery in laparoscopic resection for advanced colorectal cancer in elderly patients.
Minerva chirurgica. 2013
Madiedo A, Hall J
Minimally Invasive Management of Diverticular Disease.
Clinics in colon and rectal surgery. 2021
Olofsson F, Buchwald P, Elmståhl S, Syk I
High Tie or not in Resection for Cancer in the Sigmoid Colon?
Scand J Surg. 2018
Murata A, Amaya K, Mochizuki K, Sotokawa M, Otaka S, Tani K, Nakagaki S, Ueda T
Superior Mesenteric Artery-Pancreaticoduodenal Arcade Bypass Grafting for Repair of Inferior Pancreaticoduodenal Artery Aneurysm with Celiac Axis Occlusion.
Annals of vascular diseases. 2018
Konda S, Ihnat D, Orecchia P, Duke JM
Congenitally absent superior mesenteric artery in an asymptomatic adult.
Journal of vascular surgery cases and innovative techniques. 2021
Draginov A, Chesney TR, Quereshy HA, Chadi SA, Quereshy FA
Association of high ligation versus low ligation of the inferior mesenteric artery on anastomotic leak, postoperative complications, and mortality after minimally invasive surgery for distal sigmoid and rectal cancer.
Surgical endoscopy. 2019
Zou J, Sun Y, Yang H, Ma H, Jiang J, Jiao Y, Zhang X
Laparoscopic ligation of inferior mesenteric artery and internal iliac artery for the treatment of symptomatic type II endoleak after endovascular aneurysm repair.
International surgery. 2014
Krishnamurthy G, Menon A, Kannan K, Prakash S, Rajendran A, Philips D
Coronary artery disease and mesenteric artery stenosis - Two sides of the same coin? - Long term prospective analysis.
Intractable & rare diseases research. 2019
Turgeon MK, Gamboa AC, Regenbogen SE, Holder-Murray J, Abdel-Misih S, Hawkins AT, Silviera M, Maithel SK, Balch GC
A United States Rectal Cancer Consortium Study of Inferior Mesenteric Artery versus Superior Rectal Artery Ligation: How High Do We Need to Go?
Diseases of the colon and rectum. 2021
Matsuda K, Yokoyama S, Hotta T, Takifuji K, Watanabe T, Tamura K, Mitani Y, Iwamoto H, Mizumoto Y, Yamaue H
Oncological Outcomes following Rectal Cancer Surgery with High or Low Ligation of the Inferior Mesenteric Artery.
Gastrointestinal tumors. 2017
Famularo M, Lombardi J
Chronic Mesenteric Ischemia Presenting as Exercise-induced Abdominal Pain.
Annals of vascular surgery. 2015
Toma M, Santos A, Chess B, Muluk S, Grisafi J
The kissing-stent technique for treatment of distal aortic stenosis and protection of the inferior mesenteric artery orifice.
Journal of vascular surgery. 2012
Zang L, Ma J, Zheng M
[Effect of preserving left colic artery during radical operation of rectal cancer on anastomotic leakage and operation time].
Zhonghua wei chang wai ke za zhi = Chinese journal of gastrointestinal surgery. 2016
Hosn MA, Xu J, Sharafuddin M, Corson JD
Visceral Artery Aneurysms: Decision Making and Treatment Options in the New Era of Minimally Invasive and Endovascular Surgery.
Int J Angiol. 2019
Hu S, Li S, Huang X, Yan Y, Teng D, Lin H, He C, Gao Z, Wang Y, Du X
The effect of different inferior mesenteric artery ligation levels and different lymph node dissection areas on the short- and long-term outcomes of rectal cancer.
Journal of gastrointestinal oncology. 2021
Thum-umnuaysuk S, Boonyapibal A, Geng YY, Pattana-Arun J
Lengthening of the colon for low rectal anastomosis in a cadaveric study: how much can we gain?
Techniques in coloproctology. 2012
Hajibandeh S, Maw A
Meta-analysis and Trial Sequential Analysis of Randomized Controlled Trials Comparing High and Low Ligation of the Inferior Mesenteric Artery in Rectal Cancer Surgery.
Diseases of the colon and rectum. 2020
Cho A, Yamamoto H, Kainuma O
Tips of laparoscopic pancreaticoduodenectomy: superior mesenteric artery first approach (with video).
Journal of hepato-biliary-pancreatic sciences. 2013
Guraya SY
Optimum level of inferior mesenteric artery ligation for the left-sided colorectal cancer. Systematic review for high and low ligation continuum.
Saudi medical journal. 2016
Girard E, Trilling B, Rabattu PY, Sage PY, Taton N, Robert Y, Chaffanjon P, Faucheron JL
Level of inferior mesenteric artery ligation in low rectal cancer surgery: high tie preferred over low tie.
Techniques in coloproctology. 2019
You X, Liu Q, Wu J, Wang Y, Huang C, Cao G, Dai J, Chen D, Zhou Y
High versus low ligation of inferior mesenteric artery during laparoscopic radical resection of rectal cancer: A retrospective cohort study.
Medicine. 2020
Das Gupta J, Rana MA, Delu A, Guliani S, Langsfeld M, Marek J
Spontaneous inferior mesenteric arteriovenous fistula as a cause of severe portal hypertension and cardiomyopathy.
Journal of vascular surgery cases and innovative techniques. 2019
Yang Y, Wang G, He J, Zhang J, Xi J, Wang F
High tie versus low tie of the inferior mesenteric artery in colorectal cancer: A meta-analysis.
International journal of surgery (London, England). 2018
Mari G, Maggioni D, Costanzi A, Miranda A, Rigamonti L, Crippa J, Magistro C, Di Lernia S, Forgione A, Carnevali P, Nichelatti M, Carzaniga P, Valenti F, Rovagnati M, Berselli M, Cocozza E, Livraghi L, Origi M, Scandroglio I, Roscio F, De Luca A, Ferrari G, Pugliese R
"High or low Inferior Mesenteric Artery ligation in Laparoscopic low Anterior Resection: study protocol for a randomized controlled trial" (HIGHLOW trial).
Trials. 2015
Tsujinaka S, Kawamura YJ, Tan KY, Mizokami K, Sasaki J, Maeda T, Kuwahara Y, Konishi F, Lefor A
Proximal bowel necrosis after high ligation of the inferior mesenteric artery in colorectal surgery.
Scand J Surg. 2012
Tiselius C, Kindler C, Rosenblad A, Smedh K
Localization of mesenteric lymph node metastases in relation to the level of arterial ligation in rectal cancer surgery.
Eur J Surg Oncol. 2019
Kotsis T, Christoforou P, Nastos C, Chatziioannou A, Theodosopoulos T
Reversal of Acute Mesenteric Ischemia by Salvation of the Meandering Mesenteric Artery with Stenting of the Left Internal Iliac Artery.
Annals of vascular surgery. 2017
Li X, Li Q
[Significance of the preservation of left colic artery in laparoscopic resection of rectal cancer].
Zhonghua wei chang wai ke za zhi = Chinese journal of gastrointestinal surgery. 2018
Wang Q, Zhang C, Zhang H, Wang Y, Yuan Z, Di C
[Effect of ligation level of inferior mesenteric artery on postoperative defecation function in patients with rectal cancer].
Zhonghua wei chang wai ke za zhi = Chinese journal of gastrointestinal surgery. 2015
Robinson EC, Jordano L, Alexander JQ, Skeik N, Engstrom BI
A rare case of left colic branch aneurysm presenting with rupture and intra-abdominal hemorrhage.
Journal of vascular surgery cases and innovative techniques. 2021
Kuzu MA, Güner MA, Kocaay AF, Ä°smail E, Arslan MN, Tekdemir Ä°, Açar HÄ°
Redefining the collateral system between the superior mesenteric artery and inferior mesenteric artery: a novel classification.
Colorectal Dis. 2021
Wu Y, Peng W, Wu H, Chen G, Zhu J, Xing C
Absence of the superior mesenteric artery in an adult and a new classification method for superior-inferior mesenteric arterial variations.
Surgical and radiologic anatomy : SRA. 2013
Author Response
Thank you for the sharp and effective medline search suggested. We incorporated the article by Jolivet M. et al. and by Campbell A et al in the discussion.
Reviewer 2 Report
The authors analyzed twelve studies comparing ligation of the Arteria mesenterica inferior (AMI) and AMI-preserving technique in sigmoidectomy for diverticular diseases with regard to functional outcomes (bowel function, urinary function and sexual function; primary endpoints) as well as further secondary endpoints.
They found no significant differences regarding the primary endpoints, but a significant longer procedure time and a significant higher conversion rate.
The study is well done and the results are well interpreted.
I would recommend to make some minor revisions:
- Please list the abbreviation "RS" in table 1.
- I would delete the figure 2a, figure 2b is quite sufficient.
- May the different techniques of AMI-preserving influence the conversion rate and the procedure time? Could you perform subgroup analyses due to the technique of IMA preservation?
Author Response
Thank you for the suggestion provided.
Table 1 was corrected accordingly.
We incorporated as new sub analysis (fig 9 and 11) the requested sub analysis on the different types of IMA preservation.
Figure 2.a was deleted